# Knowledge, Attitudes, and Practices Related to Mold Remediation Following Hurricane Ida in Southeast Louisiana

**DOI:** 10.3390/ijerph21111412

**Published:** 2024-10-25

**Authors:** Anne M. Foreman, Amel Omari, Kristin J. Marks, Alyssa N. Troeschel, Emily J. Haas, Susan M. Moore, Ethan Fechter-Leggett, Ju-Hyeong Park, Jean M. Cox-Ganser, Scott A. Damon, Shannon Soileau, Colette Jacob, Arundhati Bakshi, Anna Reilly, Kathleen Aubin, Kate Puszykowski, Ginger L. Chew

**Affiliations:** 1Respiratory Health Division, National Institute for Occupational Safety and Health, Centers for Disease Control and Prevention, Morgantown, WV 26505, USA; iun8@cdc.gov (E.F.-L.); gzp8@cdc.gov (J.-H.P.); jjc8@cdc.gov (J.M.C.-G.); 2Division of Field Studies and Engineering, National Institute for Occupational Safety and Health, Centers for Disease Control and Prevention, Cincinnati, OH 45213, USA; rlx9@cdc.gov; 3Epidemic Intelligence Service, Centers for Disease Control and Prevention, Atlanta, GA 30329, USA; kma8@cdc.gov (K.J.M.); rjz5@cdc.gov (A.N.T.); 4Division of Nutrition, Physical Activity, and Obesity, National Center for Chronic Disease Prevention and Health Promotion, Deputy Director for Non-Infectious Diseases, Centers for Disease Control and Prevention, Atlanta, GA 30329, USA; 5Division of Environmental Health Science and Practice, National Center for Environmental Health, Deputy Director for Non-Infectious Diseases, Centers for Disease Control and Prevention, Atlanta, GA 30329, USA; scd3@cdc.gov (S.A.D.); gjc0@cdc.gov (G.L.C.); 6National Personal Protective Technology Laboratory, National Institute for Occupational Safety and Health, Centers for Disease Control and Prevention, Pittsburgh, PA 15236, USA; wcq3@cdc.gov (E.J.H.); sme6@cdc.gov (S.M.M.); 7Louisiana Department of Health, Baton Rouge, LA 70802, USA; shannon.soileau@la.gov (S.S.); cjacob@health.in.gov (C.J.); arundhati.bakshi@la.gov (A.B.); anna.reilly@la.gov (A.R.); kathleen.aubin@la.gov (K.A.); kate.puszykowski@la.gov (K.P.)

**Keywords:** mold, personal protective equipment, natural disasters, hurricanes

## Abstract

Hurricane Ida, a Category 4 hurricane, made landfall in southern Louisiana in August of 2021, causing widespread wind damage and flooding. The objective of this study was to investigate knowledge, attitudes, and practices related to post-hurricane mold exposure and cleanup among residents and workers in areas of Louisiana affected by Hurricane Ida and assess changes in knowledge, attitudes, and practices that have occurred over the past 16 years since Hurricane Katrina. We conducted in-person interviews with 238 residents and 68 mold-remediation workers in areas in and around New Orleans to ask about their mold cleanup knowledge and practices, personal protective equipment use, and risk perceptions related to mold. Knowledge of recommended safety measures increased since the post-Katrina survey but adherence to recommended safety measures did not. Many residents and some workers reported using insufficient personal protective equipment when cleaning up mold despite awareness of the potential negative health effects of mold exposure.

## 1. Introduction

Hurricane Ida, a Category 4 hurricane, made landfall in southern Louisiana on 29 August 2021 on the 16th anniversary of Hurricane Katrina, which hit approximately the same area [1]. With maximum sustained winds of 150 mph, Hurricane Ida caused widespread damage to homes and businesses, including heavy flooding and power outages lasting up to three weeks [2]. Ida was the sixth costliest hurricane on record in the United States, causing at least USD 75 billion in damages [3]. News media reported 91 deaths attributable to Hurricane Ida, 28 of which were in Louisiana [4]. In the weeks following the hurricane, the US Army Corps of Engineers installed over 33,600 reinforced plastic tarps on the roofs of damaged homes in 25 Louisiana parishes (a territorial division that is synonymous with “county” in other states). In Louisiana, the Federal Emergency Management Agency (FEMA) approved more than 563,500 applications for individual assistance and operated 40 Disaster Recovery Centers where residents could apply for disaster assistance, check the status of their assistance applications, and obtain other disaster-assistance-related information [5].

Flooding and water damage, combined with extended electrical outages caused by hurricanes, create warm, wet indoor environments that are ideal settings for mold growth. Living or working in damp buildings with mold has been associated with upper respiratory symptoms such as cough, wheeze, and asthma exacerbation, among other effects [6,7]. To mitigate the potential health effects of mold exposure, several federal government agencies recommend that individuals cleaning up or remediating mold should wear personal protective equipment (PPE), such as NIOSH Approved^®^ particulate air-purifying respirators (e.g., N95^®^ filtering facepiece respirators [FFRs]), gloves, and eye protection [8,9].

NIOSH Approved respirators provide the wearer with protection against the inhalation of mold (spores and fragment particles). When used correctly, NIOSH Approved N95 FFRs that properly fit the wearer seal to the face (thus termed “tight-fitting” respirators) forcing at least 90% of inhaled breath to enter through the filter media. NIOSH Approved N95 FFRs are assessed against rigorous testing, evaluation, and quality standards to ensure that at least 95% of all inhaled particulates are removed by the filter media. NIOSH Approved N95 FFRs are intended to be fit-tested prior to use to ensure the expected level of protection is achieved. NIOSH Approved N95 FFRs are also source control products, because the wearer’s exhaled breath passes through the filter media. Source control occurs when the development of a hazardous atmosphere is prevented by addressing the hazard at its source. In this context, a source control product is any face-worn product that captures the wearer’s respiratory secretions that may occur during breathing, talking, sneezing, etc.

In contrast, masks (e.g., disposable face masks and cloth masks (masks may include disposable face masks, cloth masks, or surgical masks, which serve as barriers to protect the wearer from splashes and large droplet sprays in addition to filtering particulates from the wearer’s exhaled breath)) and face coverings (e.g., bandana and neck gaiter) are intended to filter particulates from the wearer’s exhaled breath (i.e., source control), reducing the risk that those standing near the wearer will be exposed to harmful bacterial or viral particulates. While some harmful small airborne particulates will be removed during inhalation by the mask’s filter media, these filter media are not subject to the same rigorous filtration efficiency standards, resulting in a substantially lower filtration efficiency that is not sufficient to offer the same level of respiratory protection [10]. Additionally, masks do not form a seal to the wearer’s face, allowing unfiltered particulates to enter the wearer’s nose and mouth from around the edges of the mask. For these reasons, masks do not provide the same level of respiratory protection against inhalation exposure as a respirator.

In the aftermath of Hurricanes Katrina and Rita (hereafter referred to as “Hurricane Katrina”) in 2005, epidemiologists sought to understand the knowledge, attitudes, and practices (KAPs) related to mold exposure and cleanup among residents and workers in the hurricane-affected areas [11]. They found that most residents and workers understood the potential dangers of mold exposure, but most did not reliably use all recommended PPE. Since 2005, a number of events have occurred that may have changed KAPs related to mold exposure and cleaning activities. These include several hurricane and flooding events [12] and the recent coronavirus 2019 (COVID-19) pandemic during which respirators and masks were a nationwide topic of discussion. The objective of this study was to investigate the KAPs related to post-hurricane mold exposure and cleanup among residents and workers in areas of Louisiana affected by Hurricane Ida and assess changes in KAPs that have occurred over the past 16 years since Hurricane Katrina.

## 2. Materials and Methods

### 2.1. Study Population

Although Hurricane Ida caused damage in numerous states, this investigation was conducted at the request of the Louisiana Department of Health (LDH), and participation was limited to Louisiana residents. During 4–14 December 2021, we recruited participants in eight Louisiana parishes that had sustained flooding and building damage using several recruitment approaches. LDH selected these parishes based on areas identified from phone calls they received for mold/flood assistance after Hurricane Ida. Then, LDH and a major home-improvement store provided sites within those parishes at which the survey could be administered. Thirteen home improvement stores were selected as recruitment sites; these represented all of the partnered home-improvement-store locations in the selected parishes. We contacted 20 organizations conducting structural remediation after Hurricane Ida (including not-for-profit organizations and mold remediation training organizations) requesting permission to visit active remediation worksites; three organizations agreed to host our field team for worker-specific recruitment at active worksites. We also recruited participants in two farmers’ markets located in suburban areas outside of New Orleans. Finally, to recruit in hard-to-reach areas, we received permission to recruit at a FEMA Disaster Recovery Center and a health clinic, both located in rural areas heavily affected by the hurricane. Out of a total of 20 survey sites, 17 are shown in Figure 1. The three visited work sites are not included on the map to protect the confidentiality of the workers.

Participants were eligible to complete the survey if they (1) spoke English or Spanish; (2) were at least 18 years of age (the age of majority in Louisiana); and (3) reported living in an area affected by Hurricane Ida or reported working or volunteering to clean up after Hurricane Ida (hereafter referred to as workers, regardless of whether the work was completed on a volunteer basis or for wages). Verbal informed consent was obtained prior to each interview.

These survey sites and participant selection methods resulted in a convenience sample of English- or Spanish-speaking adult residents and workers in Louisiana parishes with sustained flooding and building damage from Hurricane Ida. This activity was reviewed by the Centers for Disease Control and Prevention (CDC) Institutional Review Board, deemed not research, and was conducted in accordance with applicable federal law and CDC policy (Department of Health and Human Services [HHS] Policy for Protection of Human Subjects) (See e.g., 45 C.F.R. part 46.102(l)(2); 21 C.F.R. part 56; 42 U.S.C. §241(d); 5 U.S.C. §552a; 44 U.S.C. §3501 et seq).

### 2.2. Questionnaires

We modified two related questionnaires that were administered in-person to residents and disaster recovery workers after Hurricane Katrina in 2005 [11]. The questionnaires (included in the Appendix A), one for residents and one for workers, included items related to demographics, knowledge of PPE recommendations and health effects of mold exposure, sources of information about mold, and mold-cleaning practices. If a survey participant was both a resident and a disaster recovery worker, they were administered both the resident and worker questionnaires. Most questions between the two questionnaires were identical; exceptions were six questions limited to the resident questionnaire related to the resident’s home damage and 11 occupation-related questions limited to the worker questionnaire. The questionnaires were updated from their original uses after Hurricane Katrina to account for currently available and commonly used types of respirators and masks (including face coverings such as neck gaiters and bandanas).

Terminology related to respirators and masks evolved greatly throughout the COVID-19 pandemic. At the time that the survey was administered, communications with the public commonly used the term “mask” to refer to both respirators and masks, because the primary focus for the public was “masking” for source control, which tight-fitting respirators such as the N95 FFR also provide. Therefore, the questionnaires used the term “mask” as an umbrella term for all items participants might have used with the goal of protecting their respiratory health, regardless of whether the primary protections provided by the items were respiratory protection (i.e., NIOSH Approved respirators and those meeting international standards) or source control (e.g., disposable face masks or cloth masks). Similarly, at the time this survey was administered, the term “N95” was commonly used in communications with the public to refer to any tight-fitting respirator with claims of at least 95% filtration efficiency. Therefore, the questionnaire used the term “N95” to generally refer to any type of tight-fitting respirator. While the questionnaire was administered using the umbrella term “mask”, participants viewed a display of 12 different items broadly categorized as NIOSH Approved respirators; respirators meeting international standards; and numerous source control products (Figure 2). This visual display ensured that participants would consider respirators and the full spectrum of source control products when responding, despite the simplified terminology of “mask”.

#### 2.2.1. Knowledge

To assess knowledge about how to protect against mold exposure, we asked all survey participants to look at a display of 12 different items (Figure 2) and select all that they thought they should use while cleaning up mold. Respondents could select multiple items, and we broadly categorized items for this study as NIOSH Approved respirators (cup-shaped N95 FFR [6], cup-shaped N95 FFR with valve [3], flat-fold N95 FFR [12], elastomeric half mask respirator [4], and elastomeric full facepiece respirator [5]); respirators meeting international standards (KN95 [2] and KF94 [8]); and source control products (dust [10], surgical [1], and cloth masks [7]; a neck gaiter [9]; and a bandana [11]). We also asked whether the participant thought there could still be a mold problem after all the visible mold was gone.

#### 2.2.2. Attitudes

To assess attitudes about mold and health, we asked participants if they believed that mold could make people sick and if they thought they were personally at risk of getting sick from mold. We also asked how much participants would be willing to pay for an “N95”.

#### 2.2.3. Practices

To assess mold remediation practices, PPE use during mold cleanup activities was assessed among all workers but was only assessed among residents who reported personally attempting to remove mold from their homes. These residents and workers were asked to look at the display of items (Figure 2) and select which ones they actually used to clean up mold and which ones they have at home. They could select more than one option from the display. In addition, we asked about the frequency in which participants used various PPE when cleaning mold, including “masks”, gloves, and goggles. We also asked all participants (regardless of previous mold cleanup experience) whether they had purchased or received an N95 in the last few months and what cleaning products they used or would use to clean up mold. Workers were asked additional questions about the types of PPE provided by their employer, whether they received instructions on how and when to use PPE, and how they knew that their “mask” fit.

### 2.3. Statistical Methods

Descriptive statistics, including medians, interquartile ranges, and proportions, were calculated to summarize responses to survey questions pertaining to participant characteristics and knowledge, attitudes, and practices related to post-hurricane mold cleanup activities. Fisher’s exact tests and pairwise z-tests were calculated to assess differences in proportions of participant characteristics between post-Ida residents and workers.

#### 2.3.1. Missing Data

Participants could choose not to answer any of the questions in the questionnaire. In the tables, sections with a “No response” row indicate how many participants did not answer that question.

#### 2.3.2. Post-Ida/Post-Katrina Comparison of Knowledge, Attitudes, and Practices

Fisher’s exact tests and pairwise z-tests were used to assess differences in the proportions of reported KAPs between Hurricane Katrina and Hurricane Ida samples of residents and workers. The Katrina data, including totals and proportions, were drawn from the tables in Cummings, Van Sickle [11]. For the single continuous variable (age), a two-tailed, pooled *t*-test was used to assess the difference between post-Ida and post-Katrina resident and worker samples. Descriptive statistical analyses were conducted using R (R Core Team, 2021) software.

To explore whether respondents reported different levels of recommended respirator-related knowledge, attitudes, and practices after Hurricane Ida compared with Hurricane Katrina while considering demographic differences across the two convenience samples, we used multivariable logistic regression to analyze a harmonized and concatenated dataset including both post-Ida and post-Katrina respondents. Post-Katrina data collection methods are described in a prior publication [11]. Questions pertaining to knowledge and practices were similar but not identical across the two surveys. While differences across the two surveys present limitations to the comparison analysis, we aimed to compare the extent to which participants responded to knowledge- and practice-related questions in alignment with recommendations contemporary to the time each respective survey was conducted. Appendix A describes our approach to comparing data across the two surveys.

We modeled the following three outcomes, separately for residents and workers: first, whether respondents responded in alignment with recommendations (1) or not (0) to the question “Which mask(s) should you use when cleaning up mold?” (knowledge); second, responses to the question “Do you think you are personally at risk of getting sick from mold?” (attitudes; yes vs. no/do not know); third, whether respondents wore NIOSH Approved respirators while cleaning mold (1) or not (0) (practices). Multivariable logistic regression models adjusted for gender (male, female), age (years), education (less than high school education, high school graduate, some college, college or higher graduate), race (White, Black, another or multiple races), and ethnicity (Hispanic and non-Hispanic). Adjusted odds ratios (aOR) and 95% confidence intervals (95% CI) are provided. Multivariable regressions were modeled using SAS 9.4 (SAS Institute Inc.).

## 3. Results

### 3.1. Participant Characteristics

#### Demographics and Home Damage

Of the 995 people we invited to participate after Hurricane Ida, 306 (238 residents and 68 workers) agreed to be interviewed using the questionnaire (response rate = 31%). Most participants (81%) were recruited outside of home improvement stores (Table 1).

The median age of the worker participants (49 years) was 6 years younger than that of the resident participants (55 years, *p* = 0.005). Both the resident and worker samples were majority male (60% and 72%, respectively), non-Hispanic or non-Latino (93% and 85%), and English speaking (95% and 96%). A small majority of the sampled residents were White (52%) and approximately half of the sampled workers were Black or African American (49%).

A majority of residents (69%, *n =* 164) reported home damage from Hurricane Ida. Among residents who reported home damage from Hurricane Ida, 161 provided responses regarding the amount of mold growth, 55% (88) of which reported “any amount of mold growth in their homes”.

Comparisons with Post-Katrina sample. Characteristics of the sample interviewed using the questionnaire after Hurricane Katrina are presented in Table 1 and described in a prior publication [11]. In terms of demographic comparisons with the post-Katrina resident and worker samples, there were significant differences in the racial makeup (*p* < 0.001) between the two samples. There was a lower percentage of White residents and a higher percentage of Black workers in the post-Ida sample. There were more workers with college degrees or higher in the post-Ida sample. Residents in the post-Ida sample reported less home flooding (*p* < 0.001) and smaller amounts of mold (*p* < 0.001) than the post-Katrina sample.

### 3.2. Knowledge

When shown the visual display and asked “Which mask should you use when cleaning up mold”, 188 of 238 resident participants (79%) selected only NIOSH Approved respirators that aligned with the current recommendations (Table 2). Of worker participants in the post-Ida survey, 79% (54) selected only NIOSH Approved respirators as the correct PPE to use during mold cleanup. In terms of mold knowledge, 89% of both residents and workers thought there could still be a mold problem after all the visible mold is gone. Additional information on the “mask”-related responses is included in Appendix A.

#### Comparisons with Post-Katrina Sample

In a regression model assessing respirator-related knowledge for both resident samples, adjusted for demographics (Table 3), being in the post-Ida sample (*n =* 226) was associated with higher odds of correctly selecting a NIOSH Approved respirator, compared with being in the post-Katrina sample (*n =* 155, aOR 1.78, 95% CI: 1.09–2.92). Among workers, post-Ida sample membership was not associated with correctly selecting a NIOSH Approved respirator (aOR 0.61, 95% CI: 0.18–2.05), controlling for demographic variables, though the sample size was small (nIda = 66 and nKatrina = 47).

### 3.3. Attitudes

Almost all Ida sample residents (99%; 234) and workers (99%; 66) believed that mold can make people sick (Table 2). Approximately half (48%; 113) of the residents and 64% (42) of the workers thought they were personally at risk of getting sick from mold. Similarly, 48% of residents and 66% of workers would spend greater than USD 5 for an “N95” (Table 2), which may generally be understood to mean any tight-fitting NIOSH Approved respirator and not just an N95 FFR, as discussed earlier.

Comparisons with the Post-Katrina sample: As in the post-Ida sample, most residents (96%) and workers (95%) in the post-Katrina sample believed mold can make people sick. In terms of their personal risk, a significantly smaller proportion of residents (48% vs. 66%, *p* < 0.001) and workers (64% vs. 82%, *p* = 0.02) in the post-Ida sample compared with the post-Katrina sample thought they were personally at risk of getting sick from mold.

For residents, being in the post-Ida sample was significantly associated with lower odds of thinking one was personally at risk of getting sick from mold, compared with being in the post-Katrina sample (aOR 0.46, 95% CI: 0.29–0.70) (Table 3).

The differences in reported amounts post-Ida and post-Katrina residents would pay for an “N95” were not statistically significant for residents but significant for workers. In pairwise z-tests comparing the amount workers in the two samples were willing to pay for an “N95”, a smaller proportion of workers in the post-Ida sample did not know how much they would pay (9% vs. 27%, *p* = 0.01).

### 3.4. Practices

Among the 126 post-Ida residents who personally attempted to remove mold from their home, 53% (67) reported using nothing or exclusively non-NIOSH Approved respiratory protection to protect their respiratory tract. Among these 67 residents, 43% (29) cleaned using source control products, 22% (15) used respirators meeting international standards, and 39% (26) did not wear anything to protect against inhalation of particles (Table 4). Of the 102 residents who personally attempted to remove mold from their homes and exclusively selected NIOSH Approved respirators for the knowledge question “Which mask should you use when cleaning up mold?” (Table 2), 47% (48) reported only using NIOSH Approved respirators when they cleaned up mold.

For workers, 21% (14) reported using nothing or exclusively non-NIOSH Approved respiratory protection when cleaning up mold, of which 64% (9/14) cleaned using source control products, 43% (6/14) used respirators meeting international standards, and 14% (2/14) did not wear anything to protect against the inhalation of particles. A majority (75%) of workers reported using exclusively NIOSH Approved respirators while cleaning up mold. Of the 54 workers who exclusively selected NIOSH Approved respirators for the knowledge question “Which mask should you use when cleaning up mold?” (Table 2), 81% (44) reported using only NIOSH Approved respirators when they cleaned up mold.

Of the 126 residents who reported personally attempting to remove mold from their homes, 53% (67) said they always wear a “mask”, while 20% (25) reported that they never wear a “mask” (Table 4). For workers, 75% (50) reported always wearing a “mask” while cleaning up mold and 1% (1) reported never wearing a “mask”. Additional information on “mask”-related responses is reported in Appendix A.

Fourteen (11%; 14/126) residents and zero workers reported always wearing all three types of protection (gloves, goggles, and something to protect their respiratory tract). Of the 26 workers who were employed by a company, all were provided gloves by their employer and 88% (23) were provided “masks” and goggles. Among these 26 company-employed workers, 73% (19) reported receiving instruction on how or when to use PPE and 69% (18) answered “Yes” to the question “Have you had a test to see if your work mask fits to your face?”.

In terms of what people reported using to clean up mold, 74% (170) and 70% (47) of residents and workers reported using bleach, and 42% (96) and 52% (35) selected another cleaner, most commonly a mold-specific product. Nine respondents (six residents and three workers) reported mixing bleach and ammonia. Those who reported mixing bleach and ammonia were immediately instructed on the dangers of doing so by the field team [13].

Comparisons with Post-Katrina Sample: Compared to the post-Ida sample, residents and workers in the post-Katrina sample reported similar frequencies of PPE use. In both samples, approximately half of residents and a majority of workers reported always wearing a “mask”, most residents and workers reported always wearing gloves, and a minority of residents and workers reported always wearing goggles.

After controlling for demographic covariates in the logistic regression (Table 3), being in the post-Ida sample was not significantly associated with reporting the wearing of recommended NIOSH Approved respirators during mold cleaning among residents (aOR 1.20, 95% CI: 0.59–2.41) or workers (aOR 1.27, 95% CI: 0.39–4.15).

## 4. Discussion

Among residents and workers in areas of Louisiana affected by Hurricane Ida who participated in our survey, 79% knew they should be using NIOSH Approved respirators while cleaning up mold, but in practice, only 47% were doing so. Instead, 45% reported using a (1) source control product, (2) respirator meeting international standards, or (3) nothing to protect their respiratory tract. In particular, 42% reported cleaning up mold while wearing source control products that were commonly used by the public during the COVID-19 pandemic (e.g., cloth mask and surgical mask) [14] but did not use a NIOSH Approved respirator, which was needed to protect their respiratory tract. Among residents, 43% who reported cleaning up mold did not have a NIOSH Approved respirator at home. Ensuring the availability of NIOSH Approved respirators for residents in affected areas after hurricanes might increase their adoption for mold cleanup activities. In terms of the comparisons of KAPs among resident and worker participants between post-Hurricane Katrina and post-Hurricane Ida samples, the reported practices did not differ.

There were differences, however, between the post-Katrina and post-Ida samples in resident knowledge regarding the importance of using a NIOSH Approved respirator for mold cleanup. In our study, nearly 80% of residents correctly identified NIOSH Approved respirators as what should be used during mold cleanup, which is significantly higher than in the post-Katrina study (68%) [11]. Among workers, knowledge regarding the importance of using a NIOSH Approved respirator for mold cleanup was similar across the two surveys. The differences in the residents’ responses between the two surveys might be attributable to the increased ease of access to information about appropriate PPE in the intervening 16 years for post-Ida residents versus post-Katrina residents, such as state and federal public health websites and social media, and COVID-19-related information about respirators.

Among residents and workers, frequencies of glove, goggle, respirator, or source control product use was similar in the present study and in the post-Katrina survey. After controlling for demographic variables, we did not identify significant differences across the two samples in residents’ or workers’ odds of correctly using a NIOSH Approved respirator for all mold cleanup activities with the limited sample size.

The results of this study have implications for improving the public resources and support available around respiratory protection for both workers and the general public (i.e., residents). This study’s comparison of PPE use following major hurricanes 16 years apart indicates that, although knowledge of PPE has increased, attitudes and practices toward PPE use when exposed to mold have not changed much. To this end, knowledge might be a necessary but insufficient influence on positive health behaviors [15]. Further, the depth of knowledge people have concerning the differences among types of face coverings and levels of protection offered by each type might also be a contributing factor.

Both occupational and public health interventions are needed that focus on individual knowledge regarding and attitudes toward PPE use, as well as practices around its use for workers and the general public. Knowledge must be accompanied by accessible and readily available resources that empower residents and workers to translate that knowledge into practice. As the frequency and intensity of natural disasters like hurricanes and wildfires increase [16], laborers involved in cleanup and reconstruction projects will continue to be needed in affected communities. Similarly, the number of residential property owners at risk of exposure to contaminant hazards like mold is also increasing concomitant with the increasing frequency of significant weather events [17], underscoring the growing importance of this issue.

### 4.1. PPE Practices During Mold Cleanup

In the present study, 41% of residents who cleaned up mold reported using only NIOSH Approved respirators to protect against the inhalation of mold particles, but the majority of residents reported using source control products, respirators meeting international standards, or no respiratory protection at all rather than NIOSH Approved respirators. This is consistent with some [11,18,19], but not all [20], findings from previous investigations of PPE use after hurricanes [18,19]. However, among residents of coastal New Jersey, who reported cleaning up mold after Hurricane Sandy, only 20% wore a NIOSH Approved respirator and 55% wore a “paper mask” [20]. The lower reported use of NIOSH Approved respirators for mold cleanup after Hurricane Sandy might have occurred because hurricanes occur much less frequently in New Jersey than in Louisiana, so it is possible many residents might not have appropriate PPE on hand or knowledge of why using PPE during cleanup is important. Nonetheless, 19–42% of respondents in a population-based survey reported using a source control product or a respirator when cleaning up mold, depending upon what type of cleaning agents they used (bleach being the highest percentage) [21]. The type of mold cleanup for that survey did not specify hurricane vs. everyday cleaning.

Three out of four workers in the present study reported always wearing something to protect their respiratory tract while cleaning up mold, and of those 94% wore a NIOSH Approved respirator at least some of the time. A previous assessment of Latino migrant day laborers in New Orleans found lower frequencies of use; less than half of the laborers whose jobs would require respiratory protection reported always using a dust mask or respirator [22]. Similarly, the reported occasional use of gloves (48%) and eye protection (46%) among Latino migrant day workers [22] was also lower compared with the workers in the present study. Although, the responses may be lower in the Rabido et al. study, because the participants may not have needed to wear gloves and goggles for the work they were performing. While language barriers might partially explain the lower reported use of respiratory protection during mold cleanup among Latino workers in previous studies [22], the present study’s convenience sample had a relatively low response from Hispanic (15% of workers) workers, and we were unable to conduct a stratified analysis by ethnicity.

### 4.2. Influencing Attitudes and Practices

#### 4.2.1. PPE Training and Fit-Testing Resources

Previous data have shown that many construction workers are routinely asked to assist with disaster response cleanup efforts, including after Hurricane Katrina [23,24] and likely with Ida. In many instances, construction workers do not have the health and safety training needed, including around respiratory protection [25]. Fortunately, most company-employed workers in the present study received PPE, instructions for using PPE, and a respirator fit test through their employer. In the post-Katrina survey, most company-employed workers were provided PPE and instructions for using it, and half reported having a respirator fit test [11]. However, previous studies have generally found low levels of employer health and safety support [24,26]. For example, among a random sample of 212 remediation workers in the New Orleans area after Hurricane Katrina, only 19% reported receiving PPE from their employers [24]. Additionally, in the survey of Latino day laborers, 62% of survey participants reported no safety training and 63% were provided PPE by their employers [26]. Demographic differences between these prior study samples and our convenience sample, such as the low percentage of Latino/a workers among the present study’s respondents, might have influenced discrepancies between our findings and those of previous studies.

Identifying factors associated with employer health and safety support relative to respiratory protection might be useful for guiding public health interventions that can improve personal health practices post-hurricanes. Within occupational safety and health (OSH) several organizational level factors have been identified to support worker health and safety to increase worker adherence to desired practices. Large-scale studies have identified an organization’s safety climate, which includes management values, communication, training, and safety systems, as key factors that support worker performance in the form of proactive and compliant behaviors [27]. Specific to respiratory protection, in a sample of healthcare workers, organizational safety and health support in the form of management commitment and feedback, training, and worker involvement around respiratory protection significantly predicted worker perceptions and behaviors [28]. The results of their study show that knowledge-building in the form of training and fit testing for those who are not covered by a respiratory protection plan (RPP) may benefit from health and safety support offered via their local public health departments. Moving forward, it is possible that applied interventions in community settings can help evaluate the implementation and use of respiratory protection among the public.

#### 4.2.2. Behavioral Theory and Respiratory Protection from Mold

Although the questionnaire was deployed to support a rapid public health response rather than designed to test a specific behavioral theory, the Health Belief Model (HBM) can help interpret the results and suggest future research avenues. The HBM posits that reducing perceived barriers (e.g., financial costs, effort, and time constraints) and emphasizing benefits can increase the adoption of healthier behaviors. [29]. For example, easing access to NIOSH Approved respirators could boost their use during mold remediation, as only half of the participants were willing to pay over USD 5 for one. Maintaining stockpiles to give workers who are not covered by an RPP and to the general public at no or reduced cost may increase use of respiratory protection by some people. Self-efficacy is another factor of the HBM that has been shown to predict specific health behaviors [29]. Therefore, improving not only access to NIOSH Approved respirators but also access to fit testing and training resources to improve user confidence in effective donning and doffing of respirators and other PPE [30] could also minimize challenges to behavioral adherence, along with increasing respirator efficacy for individual users. The HBM can also be used as a framework to assess individual knowledge and skills around the use of NIOSH Approved respirators. Specifically, the results suggest that the lay knowledge around NIOSH Approved respirators versus masks and other face coverings is low. Lay knowledge has been found to be a substantial determinant in estimating the health behaviors of individuals [31]. Consequently, in addition to building self-efficacy to improve confidence in use, awareness campaigns, infographics, and other easy-to-digest resources may improve lay knowledge around why properly fitting NIOSH Approved respirators such as N95 FFRs offer superior respiratory protection. Relatedly, research has shown that increased knowledge helps understand and form risk perceptions [32]. So, it is possible that outreach efforts to improve information about NIOSH Approved respirators will also influence individuals’ perceptions of risk around exposure to mold.

Capitalizing on the HBM factors of perceived susceptibility and severity may be the most critical area to address. Additionally, a previous study that applied the HBM to respiratory protection use among Malian healthcare workers found that perceived susceptibility and severity of potential illness were key factors associated with respirator use [33]. In the present study, although most residents and workers believed that mold could make people sick, they felt their personal risk of getting sick was significantly lower compared with the post-Katrina sample. Among residents, there were significantly lower odds of reporting personal risk of getting sick from mold compared to residents in the post-Katrina sample. This is suggestive that perceived susceptibility is low among contemporary residents, prompting the need to develop and implement accurate health messaging around, first, what constitutes harmful exposure to mold and, second, the effects of mold exposure on individual health. Fortunately, HBM also points toward cues to action as external signals that can increase the likelihood that individuals might make use of their knowledge in their behavioral choices. Public health interventions and messaging campaigns could serve as these cues to action that educate about the issues discussed above. If the messaging is accurate and focused, it could activate individuals’ existing knowledge regarding recommended PPE during mold remediation.

### 4.3. Limitations and Future Directions

There are several limitations to the present study. First, we obtained a convenience sample of residents and workers that is not necessarily representative of the population affected by Hurricane Ida; therefore, the findings are not generalizable beyond the study’s sample. The majority of the sample was recruited outside of home improvement stores, likely leading to the disproportionate number of homeowners in our sample (82%, compared with 67% of Louisianans [34]). Additionally, only 15% of the present worker sample identified as Hispanic or Latino, and this finding might be an underrepresentation of the population of mold remediation workers in the greater New Orleans area; a study conducted soon after Hurricane Katrina found that nearly half of remediation workers were Latino [24]. Although there were Spanish-language surveys available to potential respondents, none of the field team members were fluent Spanish speakers, which likely hindered recruitment. Another limitation is that answers to the survey questions might have been subject to social desirability bias in that respondents might not have answered all questions truthfully. For example, some respondents might have been hesitant to admit to not wearing a respirator or source control product while cleaning up mold, and the number of participants in the sample who actually wear recommended PPE when cleaning up mold might be smaller than the results indicate.

The present study found that many residents and some workers reported using insufficient (or no) PPE when cleaning up mold after Hurricane Ida despite awareness of the potential negative health effects of mold exposure. Future studies could investigate the effectiveness of focused public health messaging campaigns related to appropriate PPE selection, particularly respirators, for cleaning up mold after hurricanes and other natural disasters. This is especially relevant given the proliferation of respirators and source control products that have flooded online and physical retailers during the COVID-19 pandemic that might make the task of PPE selection for mold cleanup more difficult for consumers.

## 5. Conclusions

In conclusion, this study found that while a majority of residents and workers in areas affected by Hurricane Ida were aware of the need to use NIOSH Approved respirators for mold cleanup, actual use of NIOSH Approved respirators was low. Instead, many individuals reported using source control products, nonapproved respirators, or nothing at all to protect their respiratory tract. There were improvements in knowledge compared to a previous study conducted after Hurricane Katrina, but attitudes and practices toward PPE use remained largely unchanged. The findings suggest that interventions are needed to improve access to NIOSH Approved respirators and provide training and resources on proper PPE use. Additionally, efforts should be made to increase lay knowledge about the differences among various types of face coverings and their levels of protection, particularly in the current era in which respiratory protection measures against SARS-CoV-2 are well-advertised. Overall, both occupational and public health interventions are necessary to promote positive health behaviors and ensure the safety of individuals involved in mold cleanup activities after natural disasters.

## Figures and Tables

**Figure 1 ijerph-21-01412-f001:**
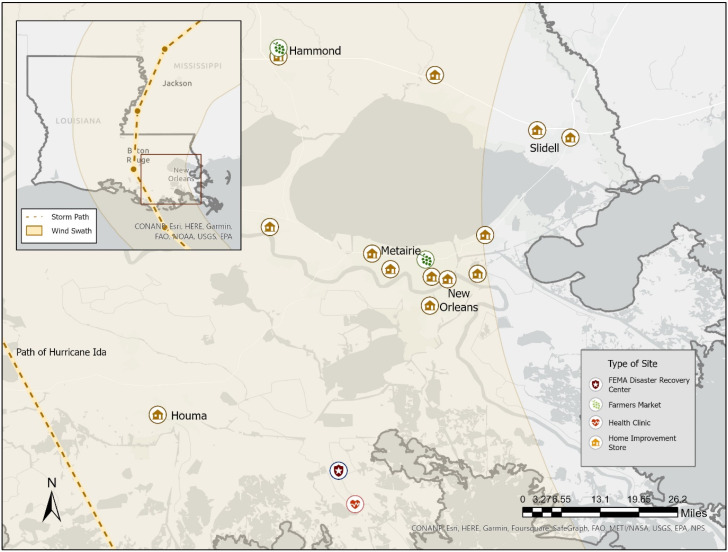
Markers indicate survey recruitment sites (*n* = 17) by type. Hurricane Ida path and range information were obtained from the National Hurricane Center. Three visited work sites are not included on the map to protect the confidentiality of the workers.

**Figure 2 ijerph-21-01412-f002:**
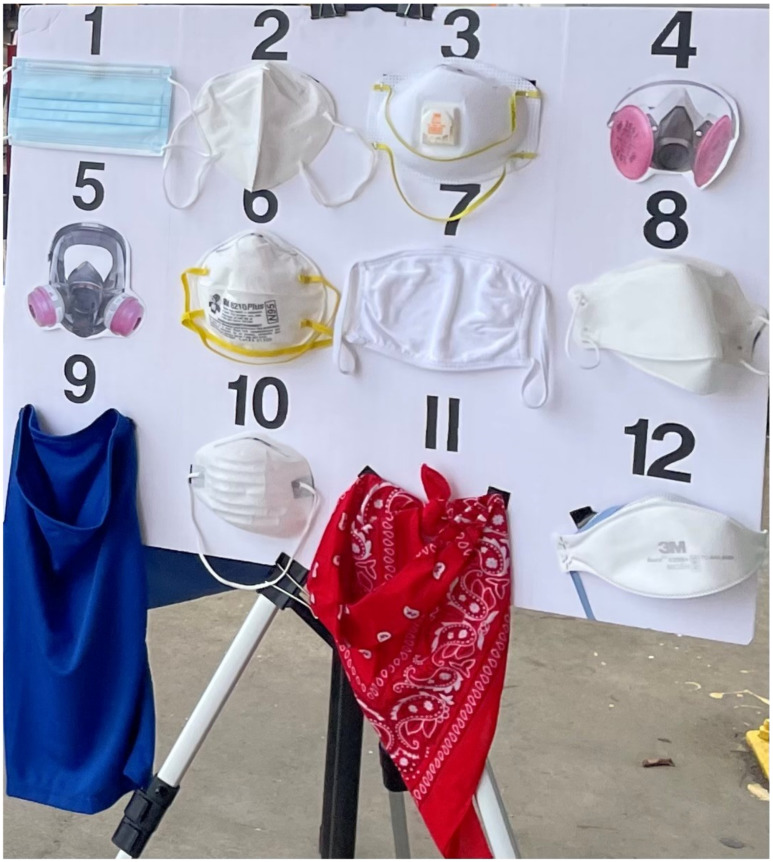
Display of items shown to participants. From the upper left, the items are (1) surgical mask; (2) Dongguan Sengtor Plastics Products Co. KN95 Particulate Disposable Respirator (Dongguan City, Guangdong Province, China); (3) 3M™ 8511 N95 Particulate Disposable Respirator with Valve (Aberdeen, SD, USA); (4) photo of a 3M™ Half Facepiece Respirator with Cartridge; (5) photo of a Honeywell 2760008A Full-Face Respirator; (6) 3M™ 8210 Plus N95 Particulate Disposable Respirator (Aberdeen, SD, USA); (7) cotton cloth face mask; (8) Kyungin Flax KF94 (Incheon, South Korea); (9) polyester neck gaiter; (10) dust mask; (11) cotton bandana; (12) 3M™9205 N95 Particulate Disposable Respirator (Aberdeen, SD, USA).

**Table 1 ijerph-21-01412-t001:** Characteristics of the surveyed residents and workers in Southeast Louisiana.

	Ida Sample			Katrina Sample		Sample Comparisons *	
Characteristic	Resident *n =* 238	Worker *n =* 68	*p*-Value *	Resident *n =* 159	Worker *n =* 76	Residents *p*-Value	Workers *p*-Value
Interview Site Type, (*n*, %)			-			-	-
Health clinic	7 (3)	0 (0)		-	-	-	-
Farmers market	35 (15)	3 (4)		-	-	-	-
FEMA Disaster Recovery Center	3 (1)	0 (0)		-	-	-	-
Home improvement store	193 (81)	52 (76)		-	-	-	-
Work site	0 (0)	13 (19)		-	-	-	-
Age, median (IQR, Range)	55 (24, 18–95)	49 (28, 21–73)	**<0.01**	51 (22, 18–81)	34 (20, 18–57)	0.26	**<0.01**
Gender, (*n*, %)			0.05			0.11	**<0.01**
Male	141 (60)	48 (72)		82 (52)	70 (92)		**<0.001**
Other gender	95 (40)	19 (28)		77 (48)	6 (8)		**<0.001**
No response	2	1		-	-		
Race, (*n*, %)			**<0.01**			**<0.001**	**<0.001**
Black or African American	73 (31)	33 (49)	**<0.001**	36 (23)	9 (19)	0.07	**<0.001**
White	121 (52)	24 (36)	**<0.001**	110 (71)	33 (70)	**<0.001**	**<0.001**
Other race	38 (16)	10 (15)	0.78	10 (6)	5 (11)	0.48	0.76
No response	6	1		3	29		
Ethnicity, (*n*, %)			0.09			-	-
Hispanic or Latino	17 (7)	10 (15)		-	-	-	-
Non-Hispanic or Latino	215 (93)	58 (85)		-	-	-	-
No response	6	0		-	-		
Primary Language, (*n*, %)			0.84			**0.01**	**<0.001**
English	223 (95)	64 (96)		158 (99)	35 (46)	-	-
No response	3	1		0	0		
Education (*n*, %)			0.59			0.19	**<0.001**
College graduate or higher	95 (41)	28 (42)		53 (33)	7 (9)		**<0.001**
Some college	67 (29)	15 (23)		45 (28)	18 (24)		0.59
High school graduate or less	70 (30)	23 (35)		61 (38)	51 (67)		**0.001**
No response	6	2		-	-		-
Type of Employment, (*n*, %)			-			-	**<0.001**
Company-employed	-	26 (38)		-	62 (82)	-	**<0.001**
Self-employed	-	28 (41)		-	14 (18)	-	**0.003**
Volunteer	-	14 (21)		-	-	-	-
No response	-	0		-	0	-	-
Damage from Hurricane, (*n*, %)			-			-	-
Yes	164 (69)	-		-	-	-	-
No	73 (31)	-		-	-	-	-
No response	1	-		-	-	-	-
Flooding			-			**<0.0001**	
Yes	36 (23)	-		107 (68)	-	-	-
No	124 (78)	-		51 (32)	-	-	-
No response	4	-		1	-		
Amount of mold			-			**<0.001**	-
None	66 (41)	-		35 (22)	-	**0.001**	-
<10 sqft	28 (17)	-		15 (9)	-	**0.04**	-
10–100 sqft	28 (17)	-		21 (13)	-	0.31	-
>100 sqft	32 (20)	-		73 (46)	-	**<0.001**	-
Do not know	7 (4)	-		-	-	-	-
No response	3	-		1	-		
Time in house with mold, Median Days (IQR, Range)	14 (98, 0–105)	-		0.5 (4, 0–42)	-	**<0.001**	-
Previous Experiences, (*n*, %)							
Prior Flooding or Mold Growth			-			**<0.001**	-
Yes	141 (59)	-		53 (33)	-	-	-
No	97 (41)	-		106 (67)	-	-	-
No response	0	-		0			

* *p*-Values centered in cells were obtained using Fisher’s exact test. *p*-Values right justified in cells were obtained using pairwise z-tests. Values less than 0.05 are bolded. Shaded cells indicate that that the results of the Fisher’s test were nonsignificant so post hoc comparisons were not performed. For the numerical variables, Age and Time in House with Mold, independent sample *t*-tests were performed.

**Table 2 ijerph-21-01412-t002:** Knowledge and attitudes about mold among surveyed residents and workers in Southeast Louisiana.

	Ida Sample		Katrina Sample		Sample Comparisons *	
Question	Resident *n =* 238	Worker *n =* 68	Resident *n =* 159	Worker *n =* 76	Resident *p*-Value	Worker *p*-Value
Knowledge Which mask(s) should you use to clean up mold? (*n*, %)					**<0.001**	0.21
Only disposable or reusable respirators (for Ida sample, restricted to only NIOSH Approved respirators)	188 (79)	54 (79)	108 (68)	65 (86)	0.02	
NIOSH Approved respirators and respirators meeting international standards	15 (6)	6 (9)	-	-	-	-
NIOSH Approved respirators and source control products	11 (5)	5 (7)	-	-	-	-
Only source control products	5 (2)	1 (1)	24 (15)	4 (5)	0.02	
Only respirators meeting international standards	4 (2)	1 (1)	-	-	-	-
NIOSH Approved respirators, respirators meeting international standards, and source control products	4 (2)	0 (0)	-	-	-	-
Respirators meeting international standards and source control products	3 (1)	0 (0)	-	-	-	-
Do not know or None	8 (3)	1 (1)	26 (16)	6 (8)	<0.001	
No response	0	0	1	0		
Attitudes						
Do you think mold can make people sick? (*n*, %)					0.06	0.52
Yes	234 (99)	66 (99)	153 (96)	72 (95)		
No	1 (<1)	1 (1)	6 (4)	4 (5)		
Do not know	1 (<1)	0 (0)	0 (0)	0 (0)		
No response	2	1	0	0		
Do you think you are personally at risk of getting sick from mold? (*n*, %)					**<0.001**	**0.02**
Yes	113 (48)	42 (64)	105 (66)	62 (82)		
No	113 (48)	23 (35)	54 (34)	14 (18)		
Do not know	10 (4)	2 (3)	0 (0)	0 (0)		
No response	2	2	0	0		
How much would you pay for an N95? (*n*, %)					0.11	**0.03**
>USD 5	111 (48)	42 (66)	90 (57)	39 (53)		0.12
USD 0–5	94 (40)	16 (25)	57 (36)	15 (20)		0.51
Do not know	28 (12)	6 (9)	11 (7)	20 (27)		**0.01**
No response	5	4	1	2		

* *p*-Values centered in cells were obtained using Fisher’s exact test. *p*-Values right justified in cells were obtained using pairwise z-tests. Values less than 0.05 are bolded. Shaded cells indicate that the results of the Fisher’s test were nonsignificant so post hoc comparisons were not performed.

**Table 3 ijerph-21-01412-t003:** Logistic regression modeling of knowledge, attitudes, and practices among residents and workers in the post-Ida versus post-Katrina samples. All models adjusted for gender, age, educational attainment, race, and ethnicity. Adjusted odds ratios and 95% confidence intervals are presented. Adjusted odds ratios with *p*-values less than 0.05 are bolded.

	Residents			Workers		
	Knowledge: Selected Recommended “Mask” aOR (95% CI) *n* = 381	Attitudes: Personally at Risk of Getting Sick from Mold aOR (95% CI) *n* = 382	Practice: Used Recommended Mask When Cleaning Mold aOR (95% CI) *n* = 174	Knowledge: Selected Recommended “Mask” aOR (95% CI) *n* = 113	Attitudes: Personally at Risk of Getting Sick from Mold aOR (95% CI) *n* = 113	Practice: Used Recommended Mask When Cleaning Mold aOR (95% CI) *n* = 99
Post-Ida respondent	**1.78** **(1.09–2.92)**	**0.46** **(0.29–0.70)**	1.20 (0.59–2.41)	0.61 (0.18–2.05)	0.79 (0.28–2.26)	1.27 (0.39–4.15)

**Table 4 ijerph-21-01412-t004:** PPE use and practices among surveyed residents and workers in Southeast Louisiana.

	Ida Sample		Katrina Sample		Sample Comparisons ^†^	
Question	Resident *n =* 126 *	Worker *n =* 68	Resident *n =* 67	Worker *n =* 69	Resident *p*-Value	Worker *p*-Value
Which mask(s) do you use to clean up mold? (*n*, %)					0.11	**<0.001**
Only respirators (for the Ida sample, restricted to only NIOSH Approved respirators)	52 (41)	51 (75)	20 (38)	43 (81)	-	0.55
Only source control products	26 (21)	6 (9)	13 (25)	3 (6)	-	0.09
Only respirators meeting international standards	12 (10)	3 (4)	-	-	-	-
Respirators meeting international standards and source control products	3 (2)	3 (4)	-	-	-	-
NIOSH Approved respirators and source control products	3 (2)	1 (1)	-	-	-	-
NIOSH Approved respirators and respirators meeting international standards	2 (2)	1 (1)	-	-	-	-
None	26 (21)	2 (3)	19 (37)	7 (13)	-	**0.03**
Do not know/Other	2 (2)	1 (1)	-	-	-	-
Which mask(s) do you have at home? *^,a^ (*n*, %)						
Source control products	91 (72)	36 (53)	-	-		
NIOSH Approved respirator	72 (57)	51 (75)	-	-		
Respirators meeting international standards	37 (29)	21 (31)	-	-		
None	11 (9)	6 (9)	-	-		
Do not know/No response	1 (<1)	0 (0)	-	-		
When cleaning mold, do you…						
Wear a mask? (*n*, %)					0.11	0.12
Always	67 (53)	50 (75)	30 (45)	47 (68)		
Often	14 (11)	7 (10)	12 (18)	10 14)		
Occasionally	20 (16)	8 (12)	5 (7)	5 (7)		
Never	25 (20)	1 (1)	19 28)	7 (10)		
Do not know/No response	0 (0)	1 (1)	1 (1)	0 (0)		
Wear gloves? (*n*, %)					0.74	0.12
Always	77 (61)	49 (73)	44 (66)	58 (84)		
Often	12 (10)	7 (10)	8 (12)	5 (7)		
Occasionally	15 (12)	10 (15)	5 (7)	3 (4)		
Never	21 (17)	1 (1)	10 (15)	3 (4)		
Do not know/No response	1 (<1)	0 (0)	0 (0)	0 (0)		
Wear goggles? (*n*, %)					0.18	0.19
Always	22 (17)	24 (36)	7 (10)	33 (48)		
Often	6 (5)	7 (10)	3 (4)	2 (3)		
Occasionally	11 (9)	15 (22)	2 (3)	11 (16)		
Never	83 (66)	21 (31)	55 (82)	21 (30)		
Do not know/No response	4 (1)	0 (0)	0 (0)	2 (3)		
Have you purchased or received an N95 in the last few months? (*n*, %)						
Yes	57 (45)	44 (65)	-	-	-	-
No	68 (54)	24 (35)	-	-	-	-
Do not know	1 (1)	0 (0)	-	-	-	-

* Of the 238 resident participants in the Ida Sample, 126 reported personally attempting to remove mold from their homes and were asked follow-up questions about PPE use and practices. ^†^ Shaded cells indicate that the results of the Fisher’s test were nonsignificant so post hoc comparisons were not performed. Values less than 0.05 are bolded. ^a^ Responses sum to more than 100% because some respondents reported having more than one type of “mask” at home.

## Data Availability

The raw data supporting the conclusions of this article will be made available by the authors upon request.

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
