# Peer review of "Knowledge, Attitudes, and Practices Related to Mold Remediation Following Hurricane Ida in Southeast Louisiana"

_ijerph, 2024, doi:10.3390/ijerph21111412_

Round 1

Reviewer 1 Report

Comments and Suggestions for Authors

Overall, knowledge about and practical use of PPE's is an important issue when handling hazardous substances, both as professionals and private residents. Therefor the results of this paper is important for planning preventive interventions for improving the use of PPE's.

The paper is well written and the design of reusing a 16 year old questionnaire used after the Katrina hurricane is fine. The researchers are up against a time sprit, where it is very difficult to recruit participants for questionnnaires, resulting in this quitte low participation rate - but difficult to change design-wise. 

Major issues:

- Compared to the relativ simple design, the Result section is far too long. With the very long tables (with clear and results and presentation) it is not nessecary to repeat so many of the results in the text. Please reduce to the highlights from the tables - and in pratice, reduce the text with at least 25%.

To a lesser degree the same goes for the Discussion. I can  be reduces somewhat by repeating less of the just presented results, and in general tightening up the wording (less words). This will improve the chance that the reader actually reads the whole article.

Besides from this it is a well prepared draft with no obious mistakes or typo's.

Author Response

Thank you for your review! In response to this reviewer's comments, we removed some redundancy from both the Results and Discussion sections. We hope you find it clearer and more straightforward.

Reviewer 2 Report

Comments and Suggestions for Authors

This study investigates knowledge, attitudes, and practices related to post-hurricane mold exposure and cleanup among residents and workers in areas of Louisiana. Major modifications include

1. survey questions should be listed as the appendix

2. the number of samples seems to be too small

3.  add some introduction on main statistical methods used

4.  a lot of p-value are missing in all tables

5.  give policy recommedation and suggest government role in such a disaster

6. the abstract and conclusion are poorly written. Main sentences in these sections can be used in any similar research. The authors should emphasize the unique finding of this study

Comments on the Quality of English Language

Minor editing of English language required

Author Response

  1. survey questions should be listed as the appendix

In the revised draft, we have added the surveys (resident and worker) to the appendix.

  1. the number of samples seems to be too small

Although we agree with the reviewer that a larger sample would have been preferable, the survey was part of a rapid public health response and thus limited by time and resources. The sample size is also similar to previously published studies that rely on data collected in similar, rapid public health response circumstances. Citations at doi: 10.15585/mmwr.mm6821a1 (n=103); doi: 10.1017/dmp.2019.6 (n=195); and doi: 10.3200/AEOH.61.3.101-108 (n=235)

We acknowledge the relatively small sample size as a limitation of the study in the Discussion section.

  1. add some introduction on main statistical methods used

The statistical methods are described in the Method section from lines 176-208.

  1. a lot of p-value are missing in all tables

We apologize for the confusion – the cells of the tables that do not have p-values either do not have enough data for the comparison to be made (e.g., if the same questions were not asked to the post-Katrina sample or vice versa) or the p-value for the Fisher’s test was not significant, so the pairwise comparisons were not made. To improve clarity, we have added dashes to the cells with no p-values due to absent data.

  1. give policy recommedation and suggest government role in such a disaster

We thank the reviewer for this comment. Although we acknowledge the important role that policy recommendations play in guiding health and safety behaviors after hurricanes and other natural disasters, it is not within our purview to give recommendations in routine scientific studies as employees within the Centers for Disease Control and Prevention. Making policy recommendations require several layers of review in the federal government and in some cases permission from Congress. The same holds for suggesting governmental roles and delineating lanes for different federal agencies.

  1. the abstract and conclusion are poorly written. Main sentences in these sections can be used in any similar research. The authors should emphasize the unique finding of this study

Revisions have been made to clarify and emphasize the unique findings of this study; specifically, emphasizing that a large proportion of respondents reported not using recommended PPE during post-hurricane mold cleanup.

Reviewer 3 Report

Comments and Suggestions for Authors

The authors have studied the knowledge, attitudes and practices (KAP)  post Ida hurricane in southern Louisiana, related to mold exposure and cleanup. The study includes residents and workers to assess their KAP in mold clean up by using a questionnaire designed to give insights about the type of PPE used, as well as their understanding of  different types  of PPE and mold exposure related health effects.  The study is relevant and almost all the aspects correlated to the study objectives is covered.  However, there are a few things that, I feel, requires attention.

·       Is there a way to know how many subjects (workers and residents) used mask, gloves and goggles all at the same time while mold clean up? This renders maximum protection and I feel it might be interesting to know that how many subjects were aware of the efficacy of using different levels of protective gears while mold cleanup.

·       In table 1, for “Race” for residents, the numbers add up to 246 ( 17+80+127+22), however the total number of recruited residents are 233. Can the authors please explain this number? Similarly for workers, the numbers add to 66. What about the remaining 1 subject?  In table 1 for education in workers numbers, the numbers add up to 62, what about the remaining 4 subjects? In table 4, for question “wear goggles” for residents, the numbers add to 123. What happened to the remaining 3 subjects ? In table 2, for “Do you think you are personally at risk of getting sick from mold?” in workers, the numbers add up to 67, although the papers say 66 subjects participated. In table 4, for the question “Which mask(s) do you have at home?”, for residents, the total number adds up to 212 (91+72+37+11+1) whereas only 126 subjects participated. Similar discrepancies are notices in the workers numbers. Transparency is crucial while reporting data, please explain the way these numbers are represented.

·       In table 1, for the question “Amount of mold”, there are 66 residents that did not have any mold in their houses. Were these same 66 residents questioned for other characteristics as well? I feel, since these residents did not have any mold, their responses to the questionnaire might be affected based on this fact.

·       Did the authors find any correlation between the education and actual usage of NIOSH approved respirators for mold clean up?

·       I feel that table S3 is an important piece of information and should be incorporated in the main manuscript.

·       Among the workers, I think it would be interesting to highlight in the results Table 2, that how many workers had prior training for using proper PPE, and how many out of those numbers used NIOSH approved respirators?

Overall, I think the study lays a direction for greater understanding to promote and implement proper safety practices, in  mold clean up. With the above modifications, I think the manuscript will be suitable for the journal.

Author Response

Thank you for your careful review of our manuscript!

  • Is there a way to know how many subjects (workers and residents) used mask, gloves and goggles all at the same time while mold clean up? This renders maximum protection and I feel it might be interesting to know that how many subjects were aware of the efficacy of using different levels of protective gears while mold cleanup.

      These findings are included in the paper: “Fourteen (11%; 14/126) residents and zero workers reported always wearing a combination of gloves, goggles, and something to protect their respiratory tract” To improve the clarity of this reported finding, we have modified it to: Fourteen (11%; 14/126) residents and zero workers reported always wearing all three types of protection (gloves, goggles, and something to protect their respiratory tract)

  • In table 1, for “Race” for residents, the numbers add up to 246 ( 17+80+127+22), however the total number of recruited residents are 233. Can the authors please explain this number? Similarly for workers, the numbers add to 66. What about the remaining 1 subject?  In table 1 for education in workers numbers, the numbers add up to 62, what about the remaining 4 subjects? In table 4, for question “wear goggles” for residents, the numbers add to 123. What happened to the remaining 3 subjects ? In table 2, for “Do you think you are personally at risk of getting sick from mold?” in workers, the numbers add up to 67, although the papers say 66 subjects participated. In table 4, for the question “Which mask(s) do you have at home?”, for residents, the total number adds up to 212 (91+72+37+11+1) whereas only 126 subjects participated. Similar discrepancies are notices in the workers numbers. Transparency is crucial while reporting data, please explain the way these numbers are represented.

      The discrepancies in the denominators are due to either 1) questions where the participants could choose more than one answer (e.g., race) or  2) missing data, which is noted in this table footnote: “† Denominators less than the total n for a column indicate missing data.” The participants had the option to answer or not answer every question, so most of the question denominators do not equal the total number of participants. It appears that the footnote symbols were very small in the manuscript table, so we have increased the font sizes of the footnote symbols to make this clearer.

  • In table 1, for the question “Amount of mold”, there are 66 residents that did not have any mold in their houses. Were these same 66 residents questioned for other characteristics as well? I feel, since these residents did not have any mold, their responses to the questionnaire might be affected based on this fact.

The residents who did not have mold in their houses were asked most of the questions in the survey. The mold cleaning practices were only asked to participants if they reported that they had personally cleaned up mold. We did conduct exploratory analyses to examine whether answers to questions did differ systematically if the participants did or did not have mold in their homes, and we did not find any significant differences (data not shown). The participants needed only to reside in a hurricane-affected area to participate in the survey, because although they may not personally have had damage from Hurricane Ida, they live in an area under constant (seasonal) threat, and so their knowledge, attitudes, and practices related to mold clean-up are relevant to the public health department.  

  • Did the authors find any correlation between the education and actual usage of NIOSH approved respirators for mold clean up?

 We did not find a systematic pattern between education and actual usage of NIOSH approved respirators for mold clean up. Here are the frequency tables for Residents and Workers (for residents, limited to those who reported personally cleaning up mold)

Residents (Fisher’s test p value = 0.0613):

No High School or Some High School

High School Graduate

Some College

College Graduate+

Chose only NIOSH Approved Respirators

7

14

9

22

52

Chose non-NIOSH Approved Respirator

5

12

27

27

71

12

26

36

49

123

  • Workers (Fisher’s test p value =0.4117):

No High School or Some High School

High School Graduate

Some College

College Graduate+

Chose only NIOSH Approved Respirators

3

15

9

23

50

Chose non-NIOSH Approved Respirator

1

4

6

5

16

4

19

15

28

66

  I feel that table S3 is an important piece of information and should be incorporated in the main manuscript.

 We agree this information is important but given the length of the paper and the density of its contents, we want to prioritize accessibility of the main findings to the reader. Information on the frequency of use of the classes of face coverings are included in the main manuscript, and the more granular information that is shown in S3 is available in the appendix.

  • Among the workers, I think it would be interesting to highlight in the results Table 2, that how many workers had prior training for using proper PPE, and how many out of those numbers used NIOSH approved respirators?

We looked at workers who did and did not have prior training for using PPE and their use of NIOSH Approved respirators and did not find a significant difference:

Fisher’s exact test p-value = 1

No Prior Training

Prior Training

Chose only NIOSH Approved Respirators

1

3

4

Chose non-NIOSH Approved Respirator

5

19

24

6

22

28

Round 2

Reviewer 2 Report

Comments and Suggestions for Authors

accepted

Author Response

Thank you for re-reviewing our manuscript!

Reviewer 3 Report

Comments and Suggestions for Authors

The discrepancies in the denominators are due to either 1) questions where the participants could choose more than one answer (e.g., race) or  2) missing data, which is noted in this table footnote: “† Denominators less than the total n for a column indicate missing data.” The participants had the option to answer or not answer every question, so most of the question denominators do not equal the total number of participants. It appears that the footnote symbols were very small in the manuscript table, so we have increased the font sizes of the footnote symbols to make this clearer.

- If there is missing data as noted in the foot note, the total number should be less than the numbers mentioned, not more. Also, "1) questions where the participants could choose more than one answer".  Do you count that as two participants? I am confused. Please explain elaborately the numbers

Author Response

Comment: - If there is missing data as noted in the foot note, the total number should be less than the numbers mentioned, not more. Also, "1) questions where the participants could choose more than one answer".  Do you count that as two participants? I am confused. Please explain elaborately the numbers

We apologize for the confusion. Our response to the question about missing values and multiple selection answers was unclear. The denominator is only less than the total number of participants (either residents or workers) in cases where there is missing data (e.g., for education, 6 residents did not answer that question, so the denominator is 232 instead of 238). 

In the case of the Race category, participants could select more than one option. When participants are given the option to select multiple responses, the sum of the numerators (the count of responses for each category) can exceed the denominator (the total number of participants). This is because each participant can contribute to multiple counts if they identify with more than one category.

For example, if a participant identifies as both Black and White, they were counted once in the "Black" category and once in the "White" category. This method can better reflect the diversity within a sample, acknowledging the complex and multifaceted nature of identity. It also respects the participants' right to self-identify in a way that best reflects their experiences and backgrounds.